# Use of Edibles as Effective Tools in Myofunctional Therapy: A Pilot Study

**DOI:** 10.3390/diagnostics14030251

**Published:** 2024-01-24

**Authors:** Sabina Saccomanno, Vincenzo Quinzi, Federica Santori, Alessia Pisaneschi, Simone Ettore Salvati, Licia Coceani Paskay, Maria Chiara Marci, Giuseppe Marzo

**Affiliations:** 1Department of Life, Health and Environmental Sciences, University of L’Aquila, 67100 L’Aquila, Italy; vincenzo.quinzi@univaq.it (V.Q.); federica.santori@student.univaq.it (F.S.); alessia.pisaneschi@student.univaq.it (A.P.); simoneettore.salvati@graduate.univaq.it (S.E.S.); mariachiara.marci@univaq.it (M.C.M.); giuseppe.marzo@univaq.it (G.M.); 2Academy of Orofacial Myofunctional Therapy (AOMT), Pacific Palisades, CA 90272, USA; lcpaskay@gmail.com

**Keywords:** deglutition disorders, myofunctional therapy, speech therapy, compliance, rehabilitation of speech

## Abstract

Aim: The aim of this retrospective study is to explore the introduction of edible spread cream and small candies as tools to improve motivation and compliance in young children undergoing myofunctional therapy, with the purpose of optimizing oral functions, including swallowing. Methods: Six patients, one female and five males, between the ages of 7 and 14 years, presenting with atypical swallowing, were evaluated and treated at the clinic of the University of L’Aquila. The patients included in the study were randomly divided into two groups and were treated with two different treatment protocols: Group A: traditional myofunctional therapy and traditional tools; Group B: same exercises as group A, but with edible tools (spreadable cream and small candies). Results: As expected, the two patients who used edible tools demonstrated increased motivation and collaboration during myofunctional therapy. Conclusions: Patient compliance, especially in very young patients, limits the effectiveness of myofunctional therapy; therefore, creative solutions are needed to achieve greater cooperation, and edible tools can play a significant part in retraining correct swallowing. Although the sample of this pilot study is small, the results suggest that using actual edible tools in myofunctional therapy could increase compliance and provide better results in myofunctional therapy.

## 1. Introduction

The craniofacial complex is a functional unity of bones, teeth, muscles, joints, glands, and the nervous system, capable of changing throughout an individual’s entire life. Two variables play a crucial role in its development and growth: the external environment and the functions that interact and influence genetic heritage [1].

Oral functions participate in the skeletal growth of young patients and are essential to facilitate harmonious development of the craniofacial complex [2]. Conversely, dentofacial dysmorphoses have a multifactorial etiology, often related to behavioral, functional, and structural abnormalities.

The origin of atypical swallowing is influenced by various factors. Indeed, its onset may be attributed to a combination of detrimental habits, environmental and hereditary factors, as well as oral and allergic diseases [3,4].

Among oral functions, atypical or dysfunctional swallowing represents one of the most frequent parafunctions, which is part of what is defined in literature as Orofacial Myofunctional Disorders (OMDs) [1,2]. Disturbances in nasal breathing may also contribute to a postural tongue thrust that persists during swallowing [5].

It is considered dysfunctional swallowing when the transition from an infant swallowing pattern to an adult pattern does not happen. In infant swallowing, there’s the physiological interposition of the tongue between the dental arches because the swallowing pattern follows the sucking pattern. The transition between the two phases takes place gradually over 12 to 15 months, along with the eruption of the deciduous teeth [6,7,8,9].

The treatment approaches offered to address atypical swallowing can be categorized as either passive (orthodontic) or active (myofunctional therapy—MFT) [8]. Orthodontic devices such as cribs, palatal spurs, and habit correctors are designed to achieve goals such as closing the anterior open bite, restoring anterior contact, and preventing dental thrust [3,10]. In contrast, myofunctional therapy (MFT) focuses on educating and addressing various functions within the stomatognathic apparatus, including swallowing, breathing, chewing, speech articulation, aesthetics, and sensory activities [4].

In a study by Ovsenik et al. [11], it appears that many children still have an atypical swallowing pattern at the age of three. That changes significantly only after the age of six in early mixed dentition, and it persists in 25% of children at the age of 12.

The establishment of OMDs, including atypical swallowing, during the skeletal growth period inevitably leads to a change in the appearance and posture of orofacial structures. The role of pediatric dentists, orthodontists, and speech therapists (among others) is crucial for early diagnosis. In this regard, multidisciplinarity is essential because these professional figures operate in the same anatomical and physiological district and aim for common results. Clinical myofunctional analysis and a customized therapy plan are essential to better understand the presence of morphological and functional alterations and how to impact their synergy [2,12]. 

In planning preventive orthodontic treatment and when screening children for orthodontic care, the clinical examination should focus on assessing incorrect orofacial functions, particularly tongue posture and function. These are primarily caused by poor sucking and feeding habits during the early stages of dental development [11,12].

While collaboration with multiple medical figures is essential for a correct diagnosis and an overall therapeutic approach, in the presence of additional health issues, the cooperation of the little patient plays a cardinal role in recovering or establishing the correct function, preventing further functional disorders [13]. It is therefore the responsibility of the dentist, the speech therapist and the parents to motivate and supervise the children in performing therapeutic exercises [12].

The clinician has several ways to evaluate the progress of myofunctional therapy, both observationally and objectively. The dynamometer is an instrument for recording the tensile force of the lips in grams and provides information regarding the maximum force with which the lips can resist the breaking of the lip seal [14]. Due to the loss of force over time of the dynamometer’s coil, but because it is still providing an immediate measurement, the dynamometer is a good motivational instrument, and its ease of use and low-tech nature makes it a good diagnostic tool in any myofunctional or dental practice. 

The Iowa Oral Performance Instrument (IOPI) is an assessment tool used to measure tongue strength and endurance [15]. The instrument was originally developed to assess relationships between tongue strength or endurance and speech motor control. Its role was later extended to investigate relationships with swallowing [15]. 

Another testing tool is fluorescein, a substance mostly used in ophthalmology and in biomarking, which can be applied on selected points on the tongue. After asking the patient to swallow, the trace can be made visible by using UV light. 

In addition, in the literature, there are several studies using a three-dimensional, noninvasive diagnostic system that allows mapping and measurement of the face’s soft tissues [16]. Such a diagnostic system is known as photogrammetry but is less suitable for small practices [17].

Regarding therapy tools, traditional treatment of orofacial myofunctional dysfunction has included little elastic bands (like the ones used in orthodontic therapy), tongue depressors, or buttons on a string, while the use of food has been traditionally limited to peanut butter, especially in the U.S.A. and before concerns for peanut allergies. Several legacy exercises in myofunctional therapy included peanut butter [4,18,19], and the most important concept that developed since the 70s is that oral myofunctional therapy needs to include more edible tools and foods much sooner in the program, as a major goal of therapy is to normalize or optimize chewing and swallowing.

Therefore, the purpose of this retrospective study is to use edible tools to evaluate whether they improve patient compliance during myofunctional therapy. For this study, only sweet foods were selected for their high acceptance rate among children. Young patients were instructed to follow an oral hygiene protocol for the duration of therapy to decrease the risk of caries [20].

Furthermore, the general health status of the young patients was assessed to mitigate the impact of sweet foods on their well-being. This is of utmost importance as childhood obesity and its associated metabolic complications continue to emerge as significant challenges in the 21st century [20,21,22,23,24].

## 2. Materials and Methods

Six patients, one female and five males, between the ages of 7 and 14 years, presenting with atypical swallowing, were evaluated and treated at the clinic of the University of L’Aquila from 2018 to 2019. All caregivers (or their parents) signed a written informed consent statement regarding the exercises of the two different protocols for myofunctional therapy. This study adhered to the Declaration of Helsinki on medical protocol and recommendations for human research. The present protocol was approved by the Ethics Committee of the University of L’Aquila (Document DR206/2013, 16 July 2013).

Patients included in our retrospective study must have observed the inclusion criteria: diagnosis of functional impairment (swallowing and oral breathing dysfunction), no posterior crossbite, no short lingual frenulum and no macroglossia, no caries experience (DMFT or dmft = 0), no food allergies and no systemic pathology (obesity). Exclusion criteria were any allergies or contraindications to the use of the indicated therapeutic products, caries experience, systemic pathology, patients with jaw contraction, macroglossia, short lingual frenulum, or other alterations that could affect the results of myofunctional treatment.

Patients included in the study were randomly divided into two groups and treated by the same myofunctional therapist. Both groups were treated with two different treatment protocols: group A with traditional myofunctional therapy with traditional tools and group B with traditional myofunctional therapy exercises but with the use of edible tools (confectionery products).

Traditional Protocol Group A:

The conventional approach typically involves a structured protocol utilizing several components, namely:The utilization of elastics and small grains of rice as tactile stimuli serves the purpose of re-educating and realigning the distorted swallowing pattern.Implementing tools such as tongue depressors or a button affixed to a cotton thread is strategically employed to enhance the strength and coordination of both the lingual and peri-oral musculature.

Edible Tools Protocol Group B:

This protocol allowed the traditional therapy tools, as listed above, to be replaced with more motivating edible tools or confectionery products. According to the literature, the confectionery products used in the various stages of therapy were hazelnut-chocolate cream spread and hard candies [25].

In particular, in the initial stages of the therapeutic course, with the aim of encouraging proprioception and sensory stimulation of the different structures of the oral cavity, we used candy to be held between the lingual apex and the retro-incisive papilla, and spreadable cream placed on the tip of the tongue, to be slid over the palate, massaging it in different directions. 

Swallowing trials were carried out with spreadable cream, which requires greater involvement of the lingual and peri-oral musculature to ensure the progression of the bolus into the pharynx, and with sweets of soft consistency, to focus on the skills of processing and preparing the bolus in the oral cavity, not requiring excessive muscular engagement.

In the final stages of treatment, after patients had mastered the individual motor patterns, the cream spread was replaced with cookies to promote masticatory muscle reinforcement and, most importantly, to generalize the re-educated swallowing function to solid consistencies as well. Tongue depressors/buttons with cotton thread sprinkled with spreadable cream or licorice sticks against which to exert strong pressure with the lingual apex were used to strengthen the lingual muscles.

Over the course of the sessions, the progressive evolution of the therapy goals and exercises was monitored according to the feedback shown by the patient and the speed of assimilation of the learned motor patterns, which absolutely varied from case to case.

At the end of each goal, patients underwent a full clinical examination at the university to evaluate the learned motor pattern. Therefore, all diagnostic tests were performed again: fluorescin testing (Payne’s technique), dynamometer measurement, photogrammetry, and photographic documentation.

The early stages of treatment aimed to achieve, through careful exploration of the oral cavity, the child’s awareness of the morphological and functional characteristics of the main structures involved in swallowing, appropriately stimulated and massaged. The patient was guided toward careful and conscious reflection on alterations in resting lingual thrust and posture in relation to the characteristics of the physiological pattern presented by the therapist. After highlighting and adequately treating any vicious habits that may be present, a period of re-education and training on the swallowing function was carried out to strengthen the deficit muscles, restore a balanced condition, and train the correct swallowing patterns.

To this end, we proceeded with a preliminary separation of the different components of the swallowing act so that they could be re-educated individually; the progression of the exercises, as illustrated in the protocols, was suggested by the muscle complexes gradually involved in swallowing. Each of these areas should, therefore, be adequately re-educated at this stage, taking into account that the muscle contractions elicited should be very large, initially slow (avoiding abrupt movements) and always regular; they should be repeated until the onset of modest fatigue; although it is recommended that the exercises be carried out daily for a total time of about 15 min, each group of contractions may be repeated several times during the day, but with an interval of at least two hours between each exercise.

Next, the individual learned motor patterns were fused together to reconstitute a single correct swallowing act. At this stage, with the aim of verifying the overall validity of the progressively rehabilitated swallowing act, foods of different consistencies were used, substituting spreadable cream with solid consistency confectionery products (cookies) in the last sessions.

The following phase is characterized by the automation of the re-educated function so that it becomes habitual and unconscious through the repetition of the exercises, according to the progression indicated by the therapist, during the performance of daily activities, such as reading or watching television, or through the use of reminders.

For this purpose and to generalize the results obtained during the therapy sessions, an exercise book was used to be given to the patients so that they could be guided to perform them to the best of their ability and with total autonomy, even in the home environment.

To prevent the use of confectionery products from causing carious lesions over time, patients were provided with an oral hygiene protocol to be performed after completion of the exercises and observed daily. The protocol included the use of a fluoride-rich mousse supplemented with daily toothpaste to prevent caries, erosions, and abrasions. Brushing teeth for at least two minutes, the use of a toothbrush with soft and or medium bristles and of appropriate size was recommended. In addition, the toothbrush was to be replaced every two to three months maximum.

Initially, all children were offered three similar hazelnut-cocoa spreadable creams to use: Nutella^®^, Novi^®^, and Consilia^®^, chosen according to similar ingredients, consistency, and respective nutritional values. 

The children did not know the name of the hazelnut-cocoa spreadable creams they were tasting. Afterward, the children were asked to choose from the three spreadable creams to their favorite. However, all children, given the choice between these three, preferred Nutella^®^, very likely due to its traditional and well-known name recognition (Table 1).

We are aware that Nutella is not a health food; however, we used it exclusively as a tool to perform the exercises, and despite the additives and palm oil in the cream, it has been shown in the literature that when taken in small doses there is no impact on the overall health of young patients [26,27,28].

Myofunctional therapy lasted a total of six months. Goals to be achieved to rehabilitate swallowing and orofacial muscles (described in Appendix A Figure A1) were set for each month. Goals for treatment procedures for both groups have been organized similarly, involving one clinic session and one home session. There was a total of 12 meetings with the myofunctional therapist, each of which lasted about 30 min, whereas the home exercise sessions included a duration of 15 min per day.

Throughout the speech therapy sessions, an exercise manual was utilized and distributed to patients, enabling independent practice within their home environment.

Myofunctional therapy began by teaching young patients the proprioception of oral cavity structures, such as the tongue and retroincisive papilla. Subsequently, exercises were introduced to correct the resting posture of the tongue. Following this, re-education was provided for the anterior, middle, and posterior parts of the tongue while also emphasizing the strengthening of the tone of the labial muscles.

During each session, the achievement of therapy objectives and the exercises were monitored according to the feedback shown by the patient, and the speed of assimilation of the motor schemes learned varied from case to case. Each therapy session also included some costal-diaphragmatic breathing exercises aimed at muscle relaxation of the various districts involved in the re-education and exercises that re-educated nasal breathing.

For each child, an instrumental diagnosis of pre- and post-myofunctional therapy was made. Extraoral and intraoral photographic records were collected. A myofunctional assessment of the lips and peri-oral muscles was made with the dynamometer, while fluorescein was used during the examination to assess the swallowing pattern. 

A diagnostic examination using photogrammetry was performed to assess soft tissues, which, through color changes on the map, allows the establishment of differences between the various parts of the face in both size and shape. Through analysis of the color variations displayed in the map, it is possible to determine the differences between the various parts of the face regarding shape and size. Each color is associated with a deviation value. The regions in green represent the area of maximum coincidence between the parts of the face (minimum deviation value), while the regions in yellow, as they progress from the lightest to the darkest shade, indicate a gradual outward departure of the overlapping part (a positive value is conventionally assigned to the deviation). Conversely, regions in light blue, in progressing from the lightest to the darkest shade, represent a progressive inward departure (a negative value is attributed in this case).

Finally, the patient’s degree of cooperation was assessed by the speech therapist himself based on an increasing score scale from 0 to 10, referring, from a qualitative-quantitative point of view to these parameters. A color code was assigned for each score (Figure 1).

## 3. Results

Pre- and post-treatment speech therapy data were collected from all children involved in the study. To immediately highlight the results obtained after myofunctional treatment and visually compare the outcomes of the two groups of treated patients, we made use of some graphs and tables shown below.

Photographic documentation shows the obtained change in lip and chin muscle tone. The improvement is confirmed by dynamometric examination, which reported an increase in lip muscle tone following myofunctional treatment (Table 2 and Table 3).

In addition, the presence of fluorescein uniquely positioned behind the retroincisive papilla and on the premolar’s palatal surface highlights the successful correction of the swallowing pattern (Table 2 and Table 3). Both Group A and Group B achieved improvements.

Photogrammetry examination shows an outward increase in the upper and lower lips, indicative of strengthening of labial muscle tone, and reduced recruitment of the facial muscles, suggesting re-education of the swallowing function (Figure 2 and Figure 3).

The average of the values showing the extent of upper and lower lip protrusion in group A and group B showed that the best results have been achieved, once again, from group B (Table 4), although the sample cannot be considered statistically significant.

Comparing the dynamometer values and the degree of cooperation in the two tables it is evident that the best results were obtained by group B, treated with sweet products.

The level of cooperation, assessed by speech therapists, was higher in patients who performed myofunctional therapy with Protocol B (Figure 4 and Figure 5), demonstrating how the use of confectionery ensured greater collaboration than the traditional protocol. 

Averaging the values testifying to the extent of upper and lower lip protrusion in group A and group B showed that the best results were once again achieved by group B, taking into account, however, that the small size of the sample does not allow for the results achieved to be considered statistically significant.

## 4. Discussion

The craniofacial structure undergoes changes over time and is distinct in that it encompasses not only bones and muscles, like other body parts but also incorporates teeth, influencing its structural development. Two key variables, genetics and function, contribute to this development. The morphofunctional balance results from the interplay of these factors. While much is still to be understood about the genetic aspect, altered function is a modifiable factor. Hence, myofunctional therapy proves to be an effective complement to orthodontics, as dysfunctional conditions can potentially lead to irreversible anomalies in facial morphology [13].

As previously highlighted in reports, the intricate relationship between form and function intricately intertwines, mutually influencing and shaping one another. Any deviation in function can significantly impact the proper development of the maxillae and dental arches, thereby exerting a profound influence on their structural configuration, impeding the potential for harmonious and balanced growth [23,29,30,31].

For example, incorrect swallowing, in addition to causing eating disorders, can alter the position of the teeth and the growth of the skeleton. Linked to swallowing is the phonatory function, in which all the muscles of the face, the tongue, and the muscles that lower and raise the jaw are employed. Lastly, respiratory function plays a crucial role because, under normal conditions, nasal passages are responsible for both inhalation and exhalation. In conditions of upper airway dysfunction, breathing will consistently occur through the mouth, creating, in turn, an altered balance of perioral and intraoral musculature, leading to structural changes in both the teeth and the skeleton. All these functions are interconnected, so an issue in any examined areas will impact the others [1].

Although swallowing is the first function to be established in the stomatognathic system, it is also the last function to be refined [7,32]. Atypical swallowing can be treated by myofunctional therapy (active treatment) and through orthodontic devices (passive treatment). Myofunctional therapy is a type of physiotherapy directed at re-educating swallowing and replace with optimizing the orofacial musculature.

A myofunctional therapy, as well as any other form of therapy involving the active participation of the patient, relies not only on modifying muscle efficiency to restore a certain function but also depends greatly on learning processes that have the power to modify neuronal circuits (neuroplasticity), sometimes temporarily and sometimes for a prolonged period [1].

Patient motivation stands as a pivotal aspect in the continuum of any enduring therapeutic process. In line with Silva et al. (2010), motivation is delineated as a complex interplay of psychological dynamics or inner energies that impel an individual toward the realization of a distinct objective. Hence, comprehending methods to incite and sustain the patient’s motivation holds paramount significance, not just within immediate time frames but more crucially across extended periods of therapeutic engagement and care [33].

Actually, in myofunctional therapy, active participation is required not only from the little patients but also from the caregivers who need to supervise therapy at home. Children must become aware of the oral cavity and its functions before undergoing myofunctional therapy in order to internalize the necessary sensory-motor functional engrams, such as tongue to palate interface at rest and in action; tongue movements within and around the oral cavity; coordination tongue, jaw, cheeks, and lips during chewing and swallowing, etc. leading up to complex functional patterns such as chewing, swallowing and speech.

One of the goals of therapy was to strengthen the muscles. For this purpose, it was necessary to separate the different components of the swallowing act so that they could be retrained individually.

Subsequently, motor patterns learned through the individual exercises were learned by the little patients, who were able to then reconstruct the swallowing act.

Particularly, this phase in Protocol B was addressed using foods of different consistencies to teach young patients the automation of the swallowing pattern so that it becomes habitual and unconscious.

Apart from research, which requires a standardized therapeutic protocol, myofunctional therapy relies on customized protocols to address the specific issues that the patient presents and to increase compliance. In this retrospective study, while positive results in myofunctional therapy were obtained in both groups, patients belonging to group B obtained higher values with the dynamometer regarding the force obtained at the lip and perioral muscles level.

The Iowa Oral Performance Instrument (IOPI) is a valuable tool for assessing the strength and endurance of the tongue and lips in patients experiencing swallowing, feeding, or speech difficulties [34,35].

Indeed, this particular device serves as a pivotal tool extensively employed across numerous scientific studies. Its primary function revolves around assessing multifaceted aspects of tongue functionality, with a specific focus on evaluating parameters such as strength and endurance, which play an integral role in formulating a customized exercise regimen tailored to each individual patient’s needs [1,36,37,38,39].

In addition, through photogrammetry, in accordance with the dynamometer results, positive values suggest an outward growth of the upper and lower lip compared with pre-therapy, indicative of increased lip muscle tone following myofunctional treatment.

This being a pilot study, it presents some limitations while suggesting several future directions. The main limitations are the small number of subjects, the non-representativeness of the population in terms of number, gender, and age, and the almost exclusive use of edible tools made of sweets. Therefore, increasing the number of subjects, lowering their age, and introducing tools made of regular foods would be imperative. Apart from edible creams made of various types of cheeses (or anything else that the child is not allergic to) for tongue to palate exercises, flat small slices of carrots or sticky creams like peanut butter or a touch of honey could be used for lip pressing, while celery sticks, carrots or almonds could be used to stimulate effective chewing. Another limitation of this pilot study was the use of a relatively common but low-tech measuring instrument, the dynamometer, which only addresses the labial function, and the use of fluorescein to trace swallowing. Future studies would need to use reliable and inexpensive tongue measuring devices such as the IOPI or, better still, Surface Electromyography (SEMG) to distinguish between an atypical swallow vs. an optimal swallow in traditional tool-based vs. edible tool-based therapy.

In the literature, there are few studies that assess long-term changes in swallowing. Among them, a particular article uses functional magnetic resonance imaging to monitor neurocerebral changes and, therefore, evaluate activations in brain areas controlling posture, tongue movements, and swallowing. While not a first-choice test, this examination can be considered [40].

## 5. Conclusions

Over the past two decades, there has been a rising clinical interest in myofunctional therapy, as evidenced by numerous papers on the subject [13]

Myofunctional therapy serves as a valuable adjunct to orthodontic treatment in cases involving detrimental habits, and when applied appropriately, it can yield favorable therapeutic outcomes. Critical to the success of the treatment are the following factors:Adherence to the home therapy regimen by the patient and their family.Collaboration among the medical team, especially when interdisciplinary treatment is necessary.Addressing associated pathologies, such as maxillary contraction, a short tongue frenum, and oral breathing resulting from adenoids and/or tonsillar hypertrophy.

Moreover, it is important to emphasize that attaining a successful therapeutic outcome is contingent upon several factors, including but not limited to precise diagnosis, prompt commencement of appropriate treatment modalities, and consistent patient engagement throughout the therapeutic process.

While acknowledging the limited size of the sample in this pilot study, the findings strongly indicate the potential benefits associated with integrating actual edible tools within myofunctional therapy. This inclusion holds promise for enhancing patient compliance and yielding superior outcomes in myofunctional therapy interventions. The challenge of patient compliance, notably among very young patients, poses a significant constraint on the efficacy of myofunctional therapy. Consequently, innovative strategies are imperative to foster heightened cooperation, and the integration of edible tools emerges as a noteworthy avenue in reestablishing proper swallowing habits.

This exploratory study observed that subjects treated with edible tools exhibited notably improved outcomes compared to those employing traditional myofunctional tools. It reiterates the value of incorporating edibles, a practice that has historical precedence from the 1970s, notably through the utilization of peanut butter. Despite children’s predilection for sweets, introducing non-sweet edibles within therapy is a viable approach to encourage appropriate chewing techniques and facilitate the widespread adoption of the correct swallowing pattern.

## Figures and Tables

**Figure 1 diagnostics-14-00251-f001:**
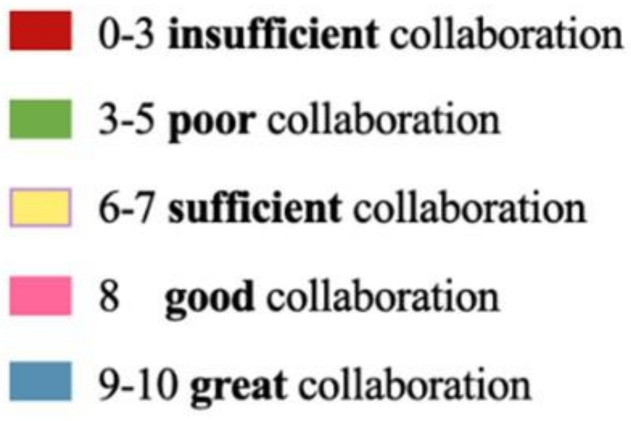
Qualitative-quantitative cooperation scale on the part of the patients in following the therapy instructions and implementing therapeutic exercises.

**Figure 2 diagnostics-14-00251-f002:**
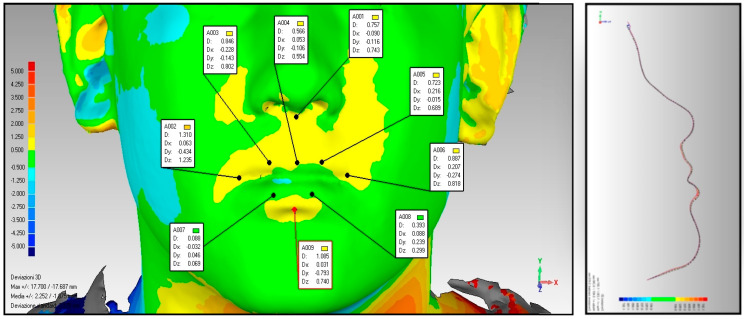
Photogrammetry of the patient P.C. belonging to group A.

**Figure 3 diagnostics-14-00251-f003:**
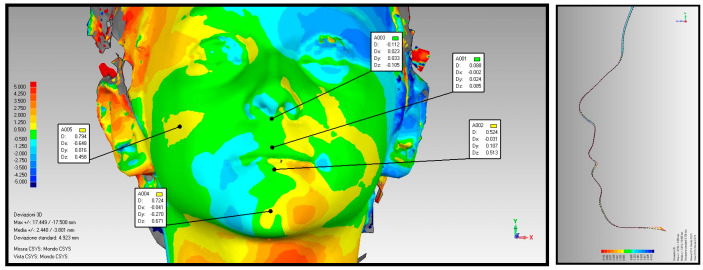
Photogrammetry of the patient B.E. belonging to group B.

**Figure 4 diagnostics-14-00251-f004:**
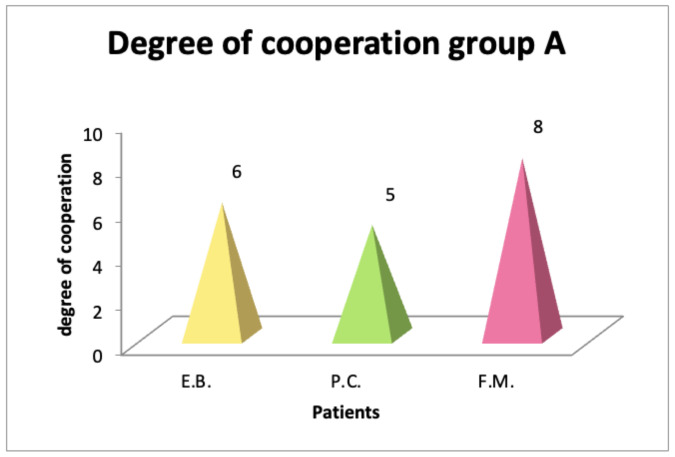
Degree of cooperation in patient group A (traditional therapy tools and exercises).

**Figure 5 diagnostics-14-00251-f005:**
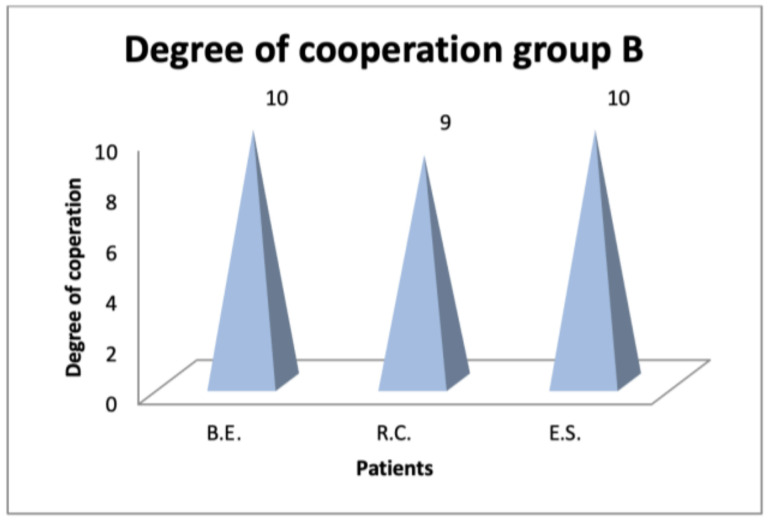
Degree of cooperation in patient group B (traditional therapy tools and exercises).

**Table 1 diagnostics-14-00251-t001:** Nutrient values for the hazelnut-cocoa spreadable cream chosen for the edible tools therapy protocol.

Nutritional Values for 100 g	Nutella Cream
Energy value	539 kcal
Protein	6.3 g
Carbohydrates	57.5
Fats	30.9

**Table 2 diagnostics-14-00251-t002:** Comparison of labial force and tongue trust pre- and post-myofunctional therapy in group A.

Group A—Traditional Protocols Therapy	Labial Muscle Strength Examined with the Dynamometer (g)		Fluorescine	
Patients	Pre	Post	Pre	Post T
E.B.	700	1000	Anterior tongue thrust	Lingual apex on retro-incisive papilla
P.C.	800	1300	Anterior tongue thrust	Lingual apex on retro-incisive papilla
F.M.	700	1100	Anterior tongue thrust	Lingual apex on retro-incisive papilla

**Table 3 diagnostics-14-00251-t003:** Comparison of labial force and tongue trust pre- and post-myofunctional therapy in group B.

Group A—Traditional Protocols Therapy	Labial Muscle Strength Examined with the Dynamometer (g)		Fluorescine	
Patients	Pre	Post	Pre	Post T
B.E.	500	2000	Anterior tongue thrust	Lingual apex on retro-incisive papilla
R.C.	600	1200	Anterior tongue thrust	Lingual apex on retro-incisive papilla
E.S.	500	1200	Anterior tongue thrust	Lingual apex on retro-incisive papilla

**Table 4 diagnostics-14-00251-t004:** Photogrammetry: Comparison of labial protrusion group A vs. group B after myofunctional therapy.

Patients						
	Group A			Group B		
	E.B.	P.C.	F.M.	B.E.	R.C.	E.S.
Protrusion of the upper lip	+0.098	+0.566	+0.881	+0.880	+2.887	+0.718
Protrusion of the lower lip	+0.077	+1.085	+0.611	+0.524	+1.748	+1.985

## Data Availability

The data presented in this study are available on request from the corresponding author.

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
