# Peer review of "Use of Edibles as Effective Tools in Myofunctional Therapy: A Pilot Study"

_diagnostics, 2024, doi:10.3390/diagnostics14030251_

Round 1
Reviewer 1 Report
Comments and Suggestions for Authors
Dear colleagues!
Thanks for the quality research, I really enjoyed the review. As some comments and clarifications, I would like to draw your attention to the following aspects
1. What was the null hypothesis of your pilot study?
2. You write about sweet foods were selected on line 109, but we know about the risk of increased caries on the background of such a “diet”. How was this aspect taken into account in the treatment?
3. On lines 205-215 you give a description of the sweet products used. In this regard, I have a number of comments and questions
a) as far as I can see in open sources, Nutella contains E 621, as well as palm oil. In addition, I would like to draw your attention to the fda warning https://www.fda.gov/safety/recalls-market-withdrawals-safety-alerts/dibe-llc-issues-allergy-alert-undeclared-soy-walnuts-and- wheat-tetas-mireya-sabor-nutella-toddy
Why did you choose such a compromised product?
b) why weren’t sweeteners or fructose content of the sweets used?
c) how did you dose sweet foods?
4. Unfortunately, I did not find high-quality assessment methods; perhaps I was not careful. Tell me, how did you qualitatively (validated tests, functional or hardware tests) evaluate the results of your rehabilitation technique?
On line 236 you indicate dynamometer and Photogrammetry (line 238) as evaluation methods, but the text does not contain information about the device and its medical affiliation, standardization, procedure for use and interpretation of the results. Have you invited an outside specialist or do you have the competencies that allow you to evaluate the results?
5. On the one hand, you indicate that you assessed Labial muscle strength examined with the dynamometer, but why then did you not use myography?
As a recommendation, I would like to advise you to collaborate with nutritionists to correctly select motivational food, as well as with functional diagnostic doctors to validly present the results of small group studies
Author Response
January 6th 2024
Dear editor,
Thank you very much for taking the time to review this manuscript. I’m sending the manuscript revised entitled “Use of edibles as effective tools in myofunctional therapy. A retrospective study”.
AUTHORS: Saccomanno S.¹, Quinzi V¹, Santori F. ¹, Pisaneschi A. ¹, Salvati S. E. ¹, Paskay Coceani L.², Marci M. C. ¹, Marzo G ¹.
¹Department of Health, Life and Environmental Science, University of L ‘Aquila, 67100 L’Aquila, Italy;
sabinasaccomanno@hotmail.it (S.S.); vincenzo.quinzi@univaq.it (V.Q.); federica.santori@student.univaq.it (F.S); alessia.pisaneschi@univaq.student.it (A.P.); simoneettore.salvati@graduate.univaq.it (S.E.S); mariachiara.marci@univaq.it (M.C.M); giuseppe.marzo@univaq.it (G.M)
2 Academy of Orofacial Myofunctional Therapy (AOMT), Pacific Palisades, CA 90272, USA;
lcpaskay@gmail.com (L.C.P)
* Correspondence: sabinasaccomanno@hotmail.it
Thank you for considering our article for publication. We were pleased to receive the generous comments of the reviewers on our manuscript. In the revised manuscript, we have carefully considered reviewers’ comments and suggestions.
The reviewer’s comments were very helpful, and we are appreciative of such constructive feedback on our submission. We added the corrections in the red in the manuscript.
Please see the attachment
We believe that the manuscript is now suitable for publication on your Journal.
Prof. ssa Sabina Saccomanno
On behalf of all the authors.
Reviewer 1
Thank you very much for taking the time to review this manuscript. Please find the detailed responses below and the corresponding corrections highlighted in red in the re-submitted files.
- Point-by-point response to Comments and Suggestions for Authors:
|
Comments 1: What was the null hypothesis of your pilot study?
|
|
Response 1: The null hypothesis of our pilot study was to evaluate whether patients' compliance level increased with the use of edibles in myofunctional therapy. Thank you for pointing this out. We agree with this comment. Therefore, we have better explained the purpose of the study on page 3, in the introduction paragraph, lines 110-111: “Therefore, the purpose of this retrospective study is to use edible tools to evaluate whether it improves patient's compliance during myofunctional therapy.”
|
|
Comments 2: You write about sweet foods were selected on line 109, but we know about the risk of increased caries on the background of such a “diet”. How was this aspect taken into account in the treatment?
|
|
Response 2: Agreed. In this regard, it was recommended that a daily home hygiene protocol be performed to prevent the occurrence of carious lesions. The protocol is described on page 5, in the description of materials and methods, lines 211-217. In addition, the sample inclusion criteria included subjects with DMFT values of zero. On page 3, in materials and methods, line 133 we describe that the exclusion criteria included patients with caries experience. We have, accordingly, modified the text to emphasize this point (page 3 and 4, lines 129-135): “Patients included in our retrospective study must observed inclusion criteria: diagnosis of functional impairment (swallowing and oral breathing dysfunction), no posterior cross bite, no short lingual frenulum and no macroglossia, no caries experience (DMFT or dmft =0), no food allergies and no systemic pathology (obesity). Exclusion criteria were any allergies or contraindications to the use of the indicated therapeutic products, caries experience, systemic pathology, patients with jaw contraction, macroglossia, short lingual frenulum, or other alterations that could affect the results of myofunctional treatment”.
Comments 3: On lines 205-215 you give a description of the sweet products used. In this regard, I have a number of comments and questions: a) as far as I can see in open sources, Nutella contains E 621, as well as palm oil. In addition, I would like to draw your attention to the fda warning https://www.fda.gov/safety/recalls-market-withdrawals-safety-alerts/dibe-llc-issues-allergy-alert-undeclared-soy-walnuts-and- wheat-tetas-mireya-sabor-nutella-toddy Why did you choose such a compromised product? b) why weren’t sweeteners or fructose content of the sweets used? c) how did you dose sweet foods? Response 3: a) We thank you for your note and would like to specify that Nutella was used only as a tool to perform the exercises and in small doses, therefore, we considered that the presence of palm oil and additives did not affect the health of the patients. A detailed literature search allowed us to find and cite the following articles: https://www.efsa.europa.eu/en/efsajournal/pub/4426; https://www.efsa.europa.eu/it/press/news/180110. In addition, the subjects considered had no food allergies or systemic diseases (page 3 and 4, line 132-134). We attach link on research on E621 stating its safety when ingested in small doses: https://www.fda.gov/food/food-additives-petitions/questions-and-answers-monosodium-glutamate-msg. We have, accordingly, revised text to highlight this point at page 5-6, lines 236-239:“We are aware that Nutella is not a health food, however, we have used it exclusively as a tool to perform the exercises, and despite the additives and palm oil in the cream, it has been shown in the literature that when taken in small doses there is no impact on the overall health of young patients”. b) We did not use sweeteners or fructose content because we needed a creamy food consistency to perform the myofunctional exercises. In confirmation of the need for a creamier consistency, the following article in the literature is cited: https://www.ncbi.nlm.nih.gov/pmc/articles/PMC8343673/. We cited this article in the text, page 4, lines 153-155. c) Thank you for pointing this out. Regarding the doses used to perform the exercises, we have integrated this information into the description of Figure 1A, on page 13, line 469-471: "For Protocol B, the doses selected for each exercise were as follows: one teaspoon of spreadable cream used during all exercises; one tic tac; one cookie to strengthen the chewing muscles after the exercises”.
Comments 4: Unfortunately, I did not find high-quality assessment methods; perhaps I was not careful. Tell me, how did you qualitatively (validated tests, functional or hardware tests) evaluate the results of your rehabilitation technique? On line 236 you indicate dynamometer and Photogrammetry (line 238) as evaluation methods, but the text does not contain information about the device and its medical affiliation, standardization, procedure for use and interpretation of the results. Have you invited an outside specialist or do you have the competencies that allow you to evaluate the results? Response 4: The results were evaluated by an external specialist: a speech therapist who followed the patients throughout the rehabilitation process and showed us the improvements. The obtained results were evaluated by using fluorescein, which showed the new position of the tongue (on the palatine spot) during the swallowing act (page 8, lines 302-304). To standardize the use of photogrammetry among operators, the following study has been taken into consideration: https://pubmed.ncbi.nlm.nih.gov/24325783/. To make the progress on result evaluation clearer, we have added the following sentence, at page 4, lines 175-178: “At the end of each goal, patients underwent a full clinical examination at the university to evaluate the learned motor pattern. Therefore, all diagnostic tests were performed again: fluorescein testing (Payne's technique), dynamometer measurement, photogrammetry and photographic documentation”.
Comments 5: On the one hand, you indicate that you assessed Labial muscle strength examined with the dynamometer, but why then did you not use myography?
Response 5: As pointed out in the text, a limitation of the study was the inability to use other instruments since only dynamometer, photogrammetry and fluorescein were present in our clinic. We hope to use other instrumentation in future studies for obtaining more reliable results.
|
|
· Response to Comments on the Quality of English Language:
|
|
Point 1: Quality of English Language (x) I am not qualified to assess the quality of English in this paper |
|
Response 1: Among the authors of the article, there is a native English-speaking colleague who has revised the text. |
Reviewer 2
Thank you for taking the time to review this manuscript. We have carefully read your comment, and we fully agree with the resulting conclusions. The study we presented aimed to introduce a different approach to myofunctional therapy in the scientific literature.
Currently, at our university, we are about to submit a research protocol to the ethics committee that will allow us to overcome the limitations of the current study. We aim to expand the sample size both in terms of numbers and in terms of gender and age diversity. However, our primary focus will be on studies involving patients in the growth phase, as collaboration seems to be a crucial element for the success of therapy.
We have considered the option of expanding the tools used in our diagnostic method, such as fluorescein, dynamometry, and photogrammetry, by acquiring instruments like iopi and surface electromyography through the university.
We are aware of recommendations against excessive candy consumption in children. However, it seems that the use of sweets in myofunctional therapy may increase children's compliance. Therefore, we plan to collaborate with a nutritionist or dietitian to create a dietary protocol suitable for these patients, avoiding potential repercussions on their overall health.
A careful assessment of personal and family medical history could further help us outline criteria for patient exclusion and inclusion.
Expanding the sample size would not only allow us to respond more accurately to the results achievable with the new therapeutic protocol but could also help us compile a list of useful foods that speech therapists worldwide could employ in their therapeutic sessions. Certainly, enlarging the sample and extending the observation period would allow us to more clearly delineate whether the results obtained through this new therapeutic protocol can have long-term effects and whether they involve changes at the cerebral level.
In accordance with the suggestions made, we found it useful to better argue the limitations of the study in the discussion section page 11, lines 408-427, as follows: “ This being a pilot study it presents some limitations while suggesting several future directions. The main limitations are the small number of subjects and the almost exclusive use of edible tools made of sweets. Therefore, increasing the number of subjects, lowering their age, and introducing tools made of regular foods would be an imperative. Apart from edible creams made of various types of cheeses (or anything else that the child is not allergic to) for tongue to palate exercises, flat small slices of carrots or sticky creams like peanut butter or a touch of honey could be used for lip pressing, while celery sticks, carrots or almonds could be used to stimulate effective chewing. Another limitation of this pilot study was the use of a relatively common but low-tech measuring instrument, the dynamometer, which only addresses the labial function; and the use of fluorescein to trace swallowing. Future studies would need to use reliable and not too expensive tongue measuring devices such as the IOPI or better still, using Surface Electromyography (SEMG) to distinguish between an atypical swallow vs. an optimal swallow in traditional tool-based vs. edible tool-based therapy. In the literature, there are few studies that assess long-term changes in swallowing. Among them, a particular article uses functional magnetic resonance imaging to monitor neurocerebral changes and, therefore, evaluate activations in brain areas controlling posture, tongue movements, and swallowing. While not a first-choice test, this examination can be considered”.
Reviewer 2 Report
Comments and Suggestions for Authors
The provided text delves into a novel approach within the realm of myofunctional therapy, specifically focusing on the use of edible tools as a means to enhance therapy outcomes for patients with atypical swallowing and other orofacial myofunctional disorders. This critical review will evaluate the study's methodology, findings, and broader implications in the context of current practices in myofunctional therapy.
**Methodology Assessment:**
The study showcases an innovative methodology by incorporating edible tools, such as sweet spreads and candies, in myofunctional therapy. This approach stands out for its creative engagement strategy, particularly aimed at pediatric patients. The use of tools like the dynamometer, fluorescein, and photogrammetry for assessment adds a layer of objective measurement to the study. However, the methodology has notable limitations, including a small sample size and a narrow focus on sweet edibles, which could raise concerns about nutritional health. The study's reliance on a limited age range and specific types of tools might also restrict the generalizability of its findings.
**Analysis of Findings:**
The results indicate a positive outcome from the use of edible tools, with improved muscle tone and swallowing patterns observed in patients. Notably, the study found increased patient cooperation in the group using edible tools compared to the group using traditional methods. This suggests that making therapy more engaging and enjoyable can significantly enhance patient compliance, a crucial factor in the success of any therapeutic intervention, especially among children.
**Critical Discussion:**
The study's exploration of the interaction between craniofacial structure and function is insightful, emphasizing the importance of early and effective intervention. It also highlights the role of patient motivation and active participation in the success of myofunctional therapy. However, the discussion could benefit from a deeper exploration of the long-term implications.
**Conclusions and Recommendations:**
The study concludes that integrating edible tools into myofunctional therapy can potentially improve patient engagement and therapeutic outcomes. While this finding is promising, the limitations of the study call for cautious interpretation. Future research should consider a larger and more diverse sample size, incorporate a variety of edible tools (including healthier options), and possibly employ more advanced diagnostic tools for a comprehensive analysis.
In conclusion, the study presents an intriguing approach to myofunctional therapy that could revolutionize treatment modalities, particularly for pediatric patients. However, its findings need to be validated through more extensive research to overcome the current limitations and fully understand the long-term impacts and efficacy of using edible tools in therapy.
Critiquing the article involves examining its various aspects critically, including the research design, methodology, scope, and conclusions. Here are some key points of critique:
1. **Sample Size and Selection Bias:**
- The study's small sample size (six patients) significantly limits the generalizability of its findings. Small sample sizes can lead to selection bias and reduce the statistical power of the study, making it difficult to draw firm conclusions.
- The age range and gender distribution (one female and five males) of the sample may not represent the broader population experiencing orofacial myofunctional disorders.
2. **Methodological Limitations:**
- The study primarily uses sweet edible tools, which raises concerns about promoting unhealthy eating habits, especially in children. This is particularly relevant in the context of rising childhood obesity and dental health issues.
- The methodology lacks a control group for a more robust comparison. Without a control group undergoing standard therapy without edible tools, it's challenging to attribute improvements solely to the use of edible tools.
3. **Potential Nutritional and Health Implications:**
- The exclusive use of sweet products, even in a therapeutic setting, might inadvertently encourage a preference for sugary foods in young patients, contradicting nutritional guidelines.
- The article does not sufficiently address how to mitigate potential negative effects, such as increased risk of dental caries or poor dietary habits, from using these sweet tools.
4. **Lack of Long-Term Follow-Up:**
- The study appears to lack a long-term follow-up to assess the sustainability of the therapeutic benefits and any potential long-term consequences (e.g., changes in dietary preferences or oral health).
5. **Generalizability of Findings:**
- The specific tools and techniques used in the study may not be easily replicable in different cultural or socio-economic contexts, limiting the applicability of the findings to a wider audience.
6. **Insufficient Data on Comparative Effectiveness:**
- The study does not provide enough comparative data to conclusively demonstrate that the edible tools are more effective than traditional myofunctional therapy tools.
7. **Ethical Considerations:**
- The study's ethical considerations, especially regarding the promotion of sweet edible tools among children, could have been explored more thoroughly. It raises questions about the responsibility of healthcare professionals in promoting healthy lifestyle choices.
In summary, while the study introduces an innovative approach to myofunctional therapy, its findings are constrained by methodological limitations, small sample size, potential health implications, and limited long-term and comparative data. These factors should be carefully considered in future research to build on this preliminary exploration effectively.
Author Response
January 6th 2024
Dear editor,
Thank you very much for taking the time to review this manuscript. I’m sending the manuscript revised entitled “Use of edibles as effective tools in myofunctional therapy. A retrospective study”.
AUTHORS: Saccomanno S.¹, Quinzi V¹, Santori F. ¹, Pisaneschi A. ¹, Salvati S. E. ¹, Paskay Coceani L.², Marci M. C. ¹, Marzo G ¹.
¹Department of Health, Life and Environmental Science, University of L ‘Aquila, 67100 L’Aquila, Italy;
sabinasaccomanno@hotmail.it (S.S.); vincenzo.quinzi@univaq.it (V.Q.); federica.santori@student.univaq.it (F.S); alessia.pisaneschi@univaq.student.it (A.P.); simoneettore.salvati@graduate.univaq.it (S.E.S); mariachiara.marci@univaq.it (M.C.M); giuseppe.marzo@univaq.it (G.M)
2 Academy of Orofacial Myofunctional Therapy (AOMT), Pacific Palisades, CA 90272, USA;
lcpaskay@gmail.com (L.C.P)
* Correspondence: sabinasaccomanno@hotmail.it
Thank you for considering our article for publication. We were pleased to receive the generous comments of the reviewers on our manuscript. In the revised manuscript, we have carefully considered reviewers’ comments and suggestions.
The reviewer’s comments were very helpful, and we are appreciative of such constructive feedback on our submission. We added the corrections in the red in the manuscript.
Please see the attachment.
We believe that the manuscript is now suitable for publication on your Journal.
Prof. ssa Sabina Saccomanno
On behalf of all the authors.
Reviewer 1
Thank you very much for taking the time to review this manuscript. Please find the detailed responses below and the corresponding corrections highlighted in red in the re-submitted files.
- Point-by-point response to Comments and Suggestions for Authors:
|
Comments 1: What was the null hypothesis of your pilot study?
|
|
Response 1: The null hypothesis of our pilot study was to evaluate whether patients' compliance level increased with the use of edibles in myofunctional therapy. Thank you for pointing this out. We agree with this comment. Therefore, we have better explained the purpose of the study on page 3, in the introduction paragraph, lines 110-111: “Therefore, the purpose of this retrospective study is to use edible tools to evaluate whether it improves patient's compliance during myofunctional therapy.”
|
|
Comments 2: You write about sweet foods were selected on line 109, but we know about the risk of increased caries on the background of such a “diet”. How was this aspect taken into account in the treatment?
|
|
Response 2: Agreed. In this regard, it was recommended that a daily home hygiene protocol be performed to prevent the occurrence of carious lesions. The protocol is described on page 5, in the description of materials and methods, lines 211-217. In addition, the sample inclusion criteria included subjects with DMFT values of zero. On page 3, in materials and methods, line 133 we describe that the exclusion criteria included patients with caries experience. We have, accordingly, modified the text to emphasize this point (page 3 and 4, lines 129-135): “Patients included in our retrospective study must observed inclusion criteria: diagnosis of functional impairment (swallowing and oral breathing dysfunction), no posterior cross bite, no short lingual frenulum and no macroglossia, no caries experience (DMFT or dmft =0), no food allergies and no systemic pathology (obesity). Exclusion criteria were any allergies or contraindications to the use of the indicated therapeutic products, caries experience, systemic pathology, patients with jaw contraction, macroglossia, short lingual frenulum, or other alterations that could affect the results of myofunctional treatment”.
Comments 3: On lines 205-215 you give a description of the sweet products used. In this regard, I have a number of comments and questions: a) as far as I can see in open sources, Nutella contains E 621, as well as palm oil. In addition, I would like to draw your attention to the fda warning https://www.fda.gov/safety/recalls-market-withdrawals-safety-alerts/dibe-llc-issues-allergy-alert-undeclared-soy-walnuts-and- wheat-tetas-mireya-sabor-nutella-toddy Why did you choose such a compromised product? b) why weren’t sweeteners or fructose content of the sweets used? c) how did you dose sweet foods? Response 3: a) We thank you for your note and would like to specify that Nutella was used only as a tool to perform the exercises and in small doses, therefore, we considered that the presence of palm oil and additives did not affect the health of the patients. A detailed literature search allowed us to find and cite the following articles: https://www.efsa.europa.eu/en/efsajournal/pub/4426; https://www.efsa.europa.eu/it/press/news/180110. In addition, the subjects considered had no food allergies or systemic diseases (page 3 and 4, line 132-134). We attach link on research on E621 stating its safety when ingested in small doses: https://www.fda.gov/food/food-additives-petitions/questions-and-answers-monosodium-glutamate-msg. We have, accordingly, revised text to highlight this point at page 5-6, lines 236-239:“We are aware that Nutella is not a health food, however, we have used it exclusively as a tool to perform the exercises, and despite the additives and palm oil in the cream, it has been shown in the literature that when taken in small doses there is no impact on the overall health of young patients”. b) We did not use sweeteners or fructose content because we needed a creamy food consistency to perform the myofunctional exercises. In confirmation of the need for a creamier consistency, the following article in the literature is cited: https://www.ncbi.nlm.nih.gov/pmc/articles/PMC8343673/. We cited this article in the text, page 4, lines 153-155. c) Thank you for pointing this out. Regarding the doses used to perform the exercises, we have integrated this information into the description of Figure 1A, on page 13, line 469-471: "For Protocol B, the doses selected for each exercise were as follows: one teaspoon of spreadable cream used during all exercises; one tic tac; one cookie to strengthen the chewing muscles after the exercises”.
Comments 4: Unfortunately, I did not find high-quality assessment methods; perhaps I was not careful. Tell me, how did you qualitatively (validated tests, functional or hardware tests) evaluate the results of your rehabilitation technique? On line 236 you indicate dynamometer and Photogrammetry (line 238) as evaluation methods, but the text does not contain information about the device and its medical affiliation, standardization, procedure for use and interpretation of the results. Have you invited an outside specialist or do you have the competencies that allow you to evaluate the results? Response 4: The results were evaluated by an external specialist: a speech therapist who followed the patients throughout the rehabilitation process and showed us the improvements. The obtained results were evaluated by using fluorescein, which showed the new position of the tongue (on the palatine spot) during the swallowing act (page 8, lines 302-304). To standardize the use of photogrammetry among operators, the following study has been taken into consideration: https://pubmed.ncbi.nlm.nih.gov/24325783/. To make the progress on result evaluation clearer, we have added the following sentence, at page 4, lines 175-178: “At the end of each goal, patients underwent a full clinical examination at the university to evaluate the learned motor pattern. Therefore, all diagnostic tests were performed again: fluorescein testing (Payne's technique), dynamometer measurement, photogrammetry and photographic documentation”.
Comments 5: On the one hand, you indicate that you assessed Labial muscle strength examined with the dynamometer, but why then did you not use myography?
Response 5: As pointed out in the text, a limitation of the study was the inability to use other instruments since only dynamometer, photogrammetry and fluorescein were present in our clinic. We hope to use other instrumentation in future studies for obtaining more reliable results.
|
|
· Response to Comments on the Quality of English Language:
|
|
Point 1: Quality of English Language (x) I am not qualified to assess the quality of English in this paper |
|
Response 1: Among the authors of the article, there is a native English-speaking colleague who has revised the text. |
Reviewer 2
Thank you for taking the time to review this manuscript. We have carefully read your comment, and we fully agree with the resulting conclusions. The study we presented aimed to introduce a different approach to myofunctional therapy in the scientific literature.
Currently, at our university, we are about to submit a research protocol to the ethics committee that will allow us to overcome the limitations of the current study. We aim to expand the sample size both in terms of numbers and in terms of gender and age diversity. However, our primary focus will be on studies involving patients in the growth phase, as collaboration seems to be a crucial element for the success of therapy.
We have considered the option of expanding the tools used in our diagnostic method, such as fluorescein, dynamometry, and photogrammetry, by acquiring instruments like iopi and surface electromyography through the university.
We are aware of recommendations against excessive candy consumption in children. However, it seems that the use of sweets in myofunctional therapy may increase children's compliance. Therefore, we plan to collaborate with a nutritionist or dietitian to create a dietary protocol suitable for these patients, avoiding potential repercussions on their overall health.
A careful assessment of personal and family medical history could further help us outline criteria for patient exclusion and inclusion.
Expanding the sample size would not only allow us to respond more accurately to the results achievable with the new therapeutic protocol but could also help us compile a list of useful foods that speech therapists worldwide could employ in their therapeutic sessions. Certainly, enlarging the sample and extending the observation period would allow us to more clearly delineate whether the results obtained through this new therapeutic protocol can have long-term effects and whether they involve changes at the cerebral level.
In accordance with the suggestions made, we found it useful to better argue the limitations of the study in the discussion section page 11, lines 408-427, as follows: “ This being a pilot study it presents some limitations while suggesting several future directions. The main limitations are the small number of subjects and the almost exclusive use of edible tools made of sweets. Therefore, increasing the number of subjects, lowering their age, and introducing tools made of regular foods would be an imperative. Apart from edible creams made of various types of cheeses (or anything else that the child is not allergic to) for tongue to palate exercises, flat small slices of carrots or sticky creams like peanut butter or a touch of honey could be used for lip pressing, while celery sticks, carrots or almonds could be used to stimulate effective chewing. Another limitation of this pilot study was the use of a relatively common but low-tech measuring instrument, the dynamometer, which only addresses the labial function; and the use of fluorescein to trace swallowing. Future studies would need to use reliable and not too expensive tongue measuring devices such as the IOPI or better still, using Surface Electromyography (SEMG) to distinguish between an atypical swallow vs. an optimal swallow in traditional tool-based vs. edible tool-based therapy. In the literature, there are few studies that assess long-term changes in swallowing. Among them, a particular article uses functional magnetic resonance imaging to monitor neurocerebral changes and, therefore, evaluate activations in brain areas controlling posture, tongue movements, and swallowing. While not a first-choice test, this examination can be considered”.